# REPRESENTATION LEARNING VIA INVARIANT CAUSAL MECHANISMS

**Jovana Mitrovic**    **Brian McWilliams**    **Jacob Walker**    **Lars Buesing**    **Charles Blundell**
DeepMind
London, UK
`{mitrovic, bmcw, jcwalker, lbuesing, cblundell}@google.com`

## ABSTRACT

Self-supervised learning has emerged as a strategy to reduce the reliance on costly supervised signals by pretraining representations only using unlabeled data. These methods combine heuristic proxy classification tasks with data augmentations and have achieved significant success, but our theoretical understanding of this success remains limited. In this paper we analyze self-supervised representation learning using a causal framework. We show how data augmentations can be more effectively utilized through explicit *invariance constraints* on the proxy classifiers employed during pretraining. Based on this, we propose a novel self-supervised objective, Representation Learning via Invariant Causal Mechanisms (RELIC), that enforces *invariant prediction* of proxy targets across augmentations through an invariance regularizer which yields improved generalization guarantees. Further, using causality we generalize contrastive learning, a particular kind of self-supervised method, and provide an alternative theoretical explanation for the success of these methods. Empirically, RELIC significantly outperforms competing methods in terms of robustness and out-of-distribution generalization on ImageNet, while also significantly outperforming these methods on Atari achieving above human-level performance on 51 out of 57 games.

## 1 INTRODUCTION

Training deep networks often relies heavily on large amounts of useful supervisory signal, such as labels for supervised learning or rewards for reinforcement learning. These training signals can be costly or otherwise impractical to acquire. On the other hand, unsupervised data is often abundantly available. Therefore, pretraining representations for unknown downstream tasks without the need for labels or extrinsic reward holds great promise for reducing the cost of applying machine learning models. To pretrain representations, self-supervised learning makes use of proxy tasks defined on unsupervised data. Recently, self-supervised methods using contrastive objectives have emerged as one of the most successful strategies for unsupervised representation learning (Oord et al., 2018; Hjelm et al., 2018; Chen et al., 2020a). These methods learn a representation by classifying every datapoint against all others datapoints (negative examples). Under assumptions on how the negative examples are sampled, minimizing the resulting contrastive loss has been justified as maximizing a lower bound on the mutual information (MI) between representations (Poole et al., 2019). However, (Tschannen et al., 2019) has shown that performance on downstream tasks may be more tightly correlated with the choice of encoder architecture than the achieved MI bound, highlighting issues with the MI theory of contrastive learning. Further, contrastive approaches compare different views of the data (usually under different data augmentations) to calculate similarity scores. This approach to computing scores has been empirically observed as a key success factor of contrastive methods, but has yet to be theoretically justified. This lack of a solid theoretical explanation for the effectiveness of contrastive methods hinders their further development.

To remedy the theoretical shortcomings, we analyze the problem of self-supervised representation learning through a causal lens. We formalize intuitions about the data generating process using a causal graph and leverage causal tools to derive properties of the optimal representation. We show that a representation should be an *invariant predictor* of proxy targets under interventions on features that are only correlated, but not causally related to the downstream targets of interest.

Since neither causally nor purely correlationally related features are observed and thus performing actual interventions on them is not feasible, for learning representation with this property we use data augmentations to simulate a subset of possible interventions. Based on our causal interpretation, we propose a regularizer which enforces that the prediction of the proxy targets is invariant across data augmentations. We propose a novel objective for self-supervised representation learning called REpresentation Learning with Invariant Causal mechanisms (RELIC). We show how this explicit invariance regularization leverages augmentations more effectively than previous self-supervised methods and that representations learned using RELIC are guaranteed to generalize well to downstream tasks under weaker assumptions than those required by previous work (Saunshi et al., 2019).

Next we generalize contrastive learning and provide an alternative theoretical explanation to MI for the success of these methods. We generalize the proxy task of instance discrimination commonly used in contrastive learning using the causal concept of *refinements* (Chalupka et al., 2014). Intuitively, a refinement of a task can be understood as a more fine-grained variant of the original problem. For example, a refinement for classifying cats against dogs would be the task of classifying individual cat and dog breeds. The instance discrimination task results from the most fine-grained refinement, e.g. discriminating individual cats and dogs from one another. We show that using refinements as proxy tasks enables us to learn useful representations for downstream tasks. Specifically, using causal tools, we show that learning a representation on refinements such that it is an invariant predictor of proxy targets across augmentations is a *sufficient condition* for these representations to generalize to downstream tasks (cf. Theorem 1). In summary, we provide theoretical support both for the general form of the contrastive objective as well as for the use of data augmentations. Thus, we provide an alternative explanation to mutual information for the success of recent contrastive approaches namely that of causal refinements of downstream tasks.

We test RELIC on a variety of prediction and reinforcement learning problems. First, we evaluate the quality of representations pretrained on ImageNet with a special focus on robustness and out-of-distribution generalization. RELIC performs competitively with current state-of-the-art methods on ImageNet, while significantly outperforming competing methods on robustness and out-of-distribution generalization of the learned representations when tested on corrupted ImageNet (ImageNet-C (Hendrycks & Dietterich, 2019)) and a version of ImageNet that consist of different renditions of the same classes (ImageNet-R (Hendrycks et al., 2020)). In terms of robustness, RELIC also significantly outperforms the supervised baseline with an absolute reduction of $4.9\%$ in error. Unlike much prior work that specifically focuses on computer vision tasks, we test RELIC for representation learning in the context of reinforcement learning on the Atari suite (Bellemare et al., 2013). There we find that RELIC significantly outperforms competing methods and achieves above human-level performance on $51$ out of 57 games.

**Contributions.**

- We formalize problem of self-supervised representation learning using causality and propose to more effectively leverage data augmentations through invariant prediction.

- We propose a new self-supervised objective, REpresentation Learning with Invariance Causal mechanisms (RELIC), that enforces invariant prediction through an explicit regularizer and show improved generalization guarantees.

- We generalize contrastive learning using refinements and show that learning on refinements is a sufficient condition for learning useful representations; this provides an alternative explanation to MI for the success of contrastive methods.

## 2  REPRESENTATION LEARNING VIA INVARIANT CAUSAL MECHANISMS

**Problem setting.**  Let $X$ denote the unlabelled observed data and $\mathcal{Y} = \{Y_t\}_{t=1}^T$ be a set of unknown tasks with $Y_t$ denoting the targets for task $t$. The tasks $\{Y_t\}_{t=1}^T$ can represent both a multi-environment as well as a multi-task setup. Our goal is to pretrain with unsupervised data a representation $f(X)$ that will be useful for solving the downstream tasks $\mathcal{Y}$.

**Causal interpretation.**  To effectively leverage common assumptions and intuitions about data generation of the unknown downstream tasks for the learning algorithm, we propose to formalize

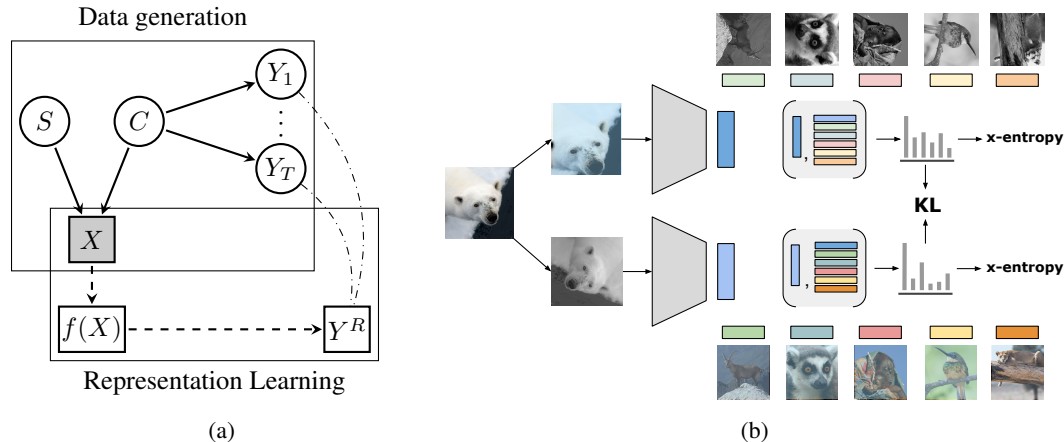

Figure 1: **(a)** Causal graph formalizing assumptions about content and style of the data and the relationship between targets and proxy tasks. The dashed arrows are not causal and represent learning, while the dashdotted lines denote that $Y^R$ is a refinement $Y_t$'s. All other arrows are causal. **(b)** RELIC objective. KL refers to the Kullback-Leibler divergence, while x-entropy denotes cross entropy.

them using a causal graph. We start from the following assumptions: a) the data is generated from *content* and *style* variables, with b) only content (and not style) being relevant for the unknown downstream tasks and c) content and style are independent, i.e. style changes are content-preserving. For example, when classifying dogs against giraffes from images, different parts of the animals constitute content, while style could be, for example, background, lighting conditions and camera lens characteristics. By assumption, content is a good representation of the data for downstream tasks and we therefore cast the goal of representation learning as estimating content. In the following, we compactly formalize these assumptions with a causal graph[1], see Figure 1a.

Let $C$ and $S$ be the latent variables describing content and style. In Figure 1a, the directed arrows from $C$ and $S$ to the observed data $X$ (e.g. images) indicate that $X$ is generated based on content and style. The directed arrow from $C$ to the target $Y_t$ (e.g. class labels) encodes the assumption that content directly influences the target tasks, while the absence of any directed arrow from $S$ to $Y_t$ indicates that style does not. Thus, content $C$ has all the necessary information to predict $Y_t$. The absence of any directed path between $C$ and $S$ in Figure 1a encodes the intuition that these variables are independent, i.e. $C \perp\!\!\!\perp S$.

Using the independence of mechanisms (Peters et al., 2017), we can conclude that under this causal model performing interventions on $S$ does not change the conditional distribution $P(Y_t|C)$, i.e. manipulating the value of $S$ does not influence this conditional distribution. Thus, $P(Y_t|C)$ is invariant under changes in style $S$. We call $C$ an *invariant representation* for $Y_t$ under $S$, i.e.

$$p^{do(S=s_i)}(Y_t \mid C) = p^{do(S=s_j)}(Y_t \mid C) \quad \forall \, s_i, s_j \in \mathcal{S}, \tag{1}$$

where $p^{do(S=s)}$ denotes the distribution arising from assigning $S$ the value $s$ with $\mathcal{S}$ the domain of $S$ (Pearl, 2009). Specifically, using $C$ as a representation allows for us to predict targets stably across perturbations, i.e. content $C$ is both a useful and robust representation for tasks $\mathcal{Y}$.

Since the targets $Y_t$ are unknown, we will construct a proxy task $Y^R$ in order to learn representations from unlabeled data $X$ only. In order to learn useful representations for $Y_t$, we will construct proxy tasks that represents more fine-grained problems than $Y_t$; for a more formal treatment of proxy tasks please refer to Section 3. Further, to learn invariant representations, such as $C$, we enforce Equation 1 which requires us to observe data under different style interventions, i.e. we need data that describes the same content under varying style. Since we do not have access to $S$, to simulate style variability we use content-preserving data augmentations (e.g. rotation, grayscaling, translation, cropping for images). Specifically, we utilize *data augmentations as interventions on the style variable $S$*, i.e.

---

[1]See (Peters et al., 2017) for a review of causal graphs and causality

applying data augmentation $a_i$ corresponds to intervening on $S$ and setting it to $s_{a_i}$. [2] Although we are not able to generate all possible styles using a fixed set of data augmentations, we will use augmentations that generate large sets of diverse styles as this allows us to learn better representations. Note that the heuristic of estimating similarity based on different views from contrastive learning can be interpreted as an implicit invariance constraint.

**RELIC objective.** Equation 1 provides a general scheme to estimate content (c.f. Figure 1a). We operationalize this by proposing to learn representations such that prediction of proxy targets from the representation is invariant under data augmentations. The representation $f(X)$ must fulfill the following *invariant prediction* criteria

$$(\textit{Invariant prediction}) \qquad p^{do(a_i)}(Y^R|f(X)) = p^{do(a_j)}(Y^R|f(X)) \quad \forall a_i, a_j \in \mathcal{A}. \qquad (2)$$

$\mathcal{A} = \{a_1, \ldots, a_m\}$ is the set of data augmentations which *simulate* interventions on the style variables and $p^{do(a)}$ denotes $p^{do(S=s_a)}$.

To achieve invariant prediction, we propose to explicitly enforce invariance under augmentations through a regularizer. This gives rise to an objective for self-supervised learning we call Representation Learning via Invariant Causal Mechanisms (RELIC). We write this objective as

$$\min_{X \sim p(X)} \mathbb{E}_{\substack{a_{lk}, a_{qt} \\ \sim \mathcal{A} \times \mathcal{A}}} \sum_{b \in \{a_{lk}, a_{qt}\}} \mathcal{L}_b(Y^R, f(X))$$

$$s.t. \quad KL\left(p^{do(a_{lk})}(Y^R|\, f(X)), p^{do(a_{qt})}(Y^R|\, f(X))\right) \leq \rho$$

where $\mathcal{L}$ is the proxy task loss and $KL$ is the Kullback-Leibler (KL) divergence. Note that any distance measure on distributions can be used in place of the KL divergence. We explain the remaining terms in detail below.

Concretely, as proxy task we associate to every datapoint $x_i$ the label $y_i^R = i$. This corresponds to the instance discrimination task, commonly used in contrastive learning (Hadsell et al., 2006). We take pairs of points $(x_i, x_j)$ to compute similarity scores and use pairs of augmentations $a_{lk} = (a_l, a_k) \in \mathcal{A} \times \mathcal{A}$ to perform a style intervention. Given a batch of samples $\{x_i\}_{i=1}^N \sim \mathcal{D}$, we use

$$p^{do(a_{lk})}(Y^R = j \,|\, f(x_i)) \propto \exp\left(\phi(f(x_i^{a_l}), h(x_j^{a_k}))/\tau\right).$$

with $x^a$ data augmented with $a$ and $\tau$ a softmax temperature parameter. We encode $f$ using a neural network and choose $h$ to be related to $f$, e.g. $h = f$ or as a network with an exponential moving average of the weights of $f$ (e.g. target networks similar to (Grill et al., 2020)). To compare representations we use the function $\phi(f(x_i), h(x_j)) = \langle g(f(x_i)), g(h(x_j)) \rangle$ where $g$ is a fully-connected neural network often called the critic.

Combining these pieces, we learn representations by minimizing the following objective over the full set of data $x_i \in \mathcal{D}$ and augmentations $a_{lk} \in \mathcal{A} \times \mathcal{A}$

$$\min -\sum_{i=1}^N \sum_{a_{lk}} \log \frac{\exp\left(\phi(f(x_i^{a_l}), h(x_i^{a_k}))/\tau\right)}{\sum_{m=1}^M \exp\left(\phi(f(x_i^{a_l}), h(x_m^{a_k}))/\tau\right)} + \alpha \sum_{a_{lk}, a_{qt}} KL(p^{do(a_{lk})}, p^{do(a_{qt})}) \qquad (3)$$

with $M$ the number of points we use to construct the contrast set and $\alpha$ the weighting of the invariance penalty. We used the shorthand $p^{do(a)}$ for $p^{do(a)}(Y^R = j \,|\, f(x_i))$. With appropriate choices for $\phi$, $g$, $f$ and $h$ above, Equation 3 recovers many recent state-of-the-art methods (c.f. Table 5 in Section A). Figure 1b presents a schematic of the RELIC objective.

The explicit invariance penalty encourages the within-class distances (for a downstream task of interest) of the representations learned by RELIC to be tightly concentrated. We show this empirically in Figure 2 and theoretically in Appendix B. In the following section we provide theoretical justification for using an instance discrimination-based contrastive loss using a causal perspective. We also show (cf. Theorem 1 below) that minimizing the contrastive loss alone (i.e. $\alpha = 0$) does not guarantee generalization. Instead, invariance across augmentations must be explicitly enforced.

---

[2]Since neither content nor style are a priori known, choosing a set of augmentations implicitly defines which aspects of the data are considered style and which are content.

## 3 GENERALIZING CONTRASTIVE LEARNING

**Learning with refinements.** In contrastive learning, the task of instance discrimination, i.e. classifying the dataset $\{(x_i, y_i^R = i) | x_i \in \mathcal{D}\}$, is used as the proxy task. To better understand contrastive learning and motivate this proxy task, we generalize instance discrimination using the causal concept of *refinements* (Chalupka et al., 2014). Intuitively, a refinement of one problem is another more fine-grained problem. If task $Y_t$ is to classify cats against dogs, then a refinement of $Y_t$ is the task of classifying cats and dogs into their individual breeds. See Figure 4 for a further visual example. For any set of tasks, there exist many different refinements. However, the most fine-grained refinement corresponds exactly to classifying the dataset $\{(x_i, y_i^R = i) | x_i \in \mathcal{D}\}$. Thus, the instance discrimination task used in contrastive learning is a specific type of refinement. For a definition and formal treatment of refinements please refer to Appendix D.

Let $Y^R$ be targets of a proxy task that is a refinement for all tasks in $\mathcal{Y}$. Leveraging causal tools, we connect learning on refinements to learning on downstream tasks. Specifically, we provide a theoretical justification for exchanging unknown downstream tasks with these specially constructed proxy tasks. We show that if $f(X)$ is an invariant representation for $Y^R$ under changes in style $S$, then $f(X)$ is also an invariant representation for tasks in $\mathcal{Y}$ under changes in style $S$. Thus by enforcing invariance under style interventions on a refinement, we learn representations that generalize to downstream tasks.[3] This is summarized in the following theorem.

**Theorem 1.** *Let $\mathcal{Y} = \{Y_t\}_{t=1}^T$ be a family of downstream tasks. Let $Y^R$ be a refinement for all tasks in $\mathcal{Y}$. If $f(X)$ is an invariant representation for $Y^R$ under style interventions $S$, then $f(X)$ is an invariant representation for all tasks in $\mathcal{Y}$ under style interventions $S$, i.e.*

$$p^{do(s_i)}(Y^R \,|\, f(X)) = p^{do(s_j)}(Y^R \,|\, f(X)) \quad \Rightarrow \quad p^{do(s_i)}(Y_t \,|\, f(X)) = p^{do(s_j)}(Y_t \,|\, f(X)) \quad (4)$$

*for all $s_i, s_j \in \mathcal{S}$ with $p^{do(s_i)} = p^{do(S=s_i)}$. Thus, $f(X)$ is a representation that generalizes to $\mathcal{Y}$.*

Theorem 1 states that if $Y^R$ is a refinement of $\mathcal{Y}$ then learning a representation on $Y^R$ is a *sufficient* condition for this representation to be useful on $\mathcal{Y}$. For a formal exposition of these points and accompanying proofs, please refer to Appendix D. Recall that the instance discrimination proxy task is the most fine-grained refinement, and so the right hand side of 4 is satisfied for any downstream task satisfying the stated assumptions of the theorem.

We generalize contrastive learning through refinements and connect representations learned on refinements and downstream tasks in Theorem 1. Thus, using causality we provide an alternative explanation to mutual information for the success of contrastive learning. Note that our methodology of refinements is not limited to instance discrimination tasks and is thus more general than currently used contrastive losses. Real world data often includes rich sources of metadata which can be used to guide the construction of refinements by grouping the data according to any available meta-data. Note that the coarser we can create a refinement, the more data efficient we can expect to be when learning representations for downstream tasks. Further, we can also expect to require less supervised data to finetune the representation.

## 4 RELATED WORK

**Contrastive objectives and mutual information maximization.** Many recent approaches to self-supervised learning are rooted in the well-established idea of maximizing mutual information (MI), e.g. Contrastive Predictive Coding (CPC) (Oord et al., 2018; Hénaff et al., 2019), Deep InfoMax (DIM) (Hjelm et al., 2018) and Augmented Multiscale DIM (AMDIM) (Bachman et al., 2019). These methods are based on noise contrastive estimation (NCE) (Gutmann & Hyvärinen, 2010) which, under specific conditions, can be viewed as a bound on MI (Poole et al., 2019). The resulting objective functions are commonly referred to as InfoNCE.

The precise role played by mutual information maximization in self-supervised learning is subject to some debate. (Tschannen et al., 2019) argue that the performance on downstream tasks is

---

[3]Note that since refinements are more fine-grained that the original task, if a representation captures a refinement then it also captures the downstream tasks as strictly more information is needed to solve the refinement.

not correlated with the achieved bound on MI, but may be more tightly correlated with encoder architecture and capacity. Importantly, InfoNCE objectives require custom architectures to ensure the network does not converge to non-informative solutions thus precluding the use of standard architectures. Recently, several works (He et al., 2019; Chen et al., 2020a) successfully combined contrastive estimation with a standard ResNet-50 architecture. In particular, SimCLR (Chen et al., 2020a) relies on a set of *strong* augmentations[4], while (He et al., 2019) uses a memory bank. Inspired by target networks in reinforcement learning, (Grill et al., 2020) proposed BYOL: an algorithm for self-supervised learning which remarkably does not use a contrastive objective. Although theoretical explanation for the good performance of BYOL is presently missing, interestingly the objective, an $\ell_2$ distance between two different embeddings of the input data resembles the $\ell_2$ form of our regularizer proposed in Equation 5 in Appendix B.

Recently, (Saunshi et al., 2019) proposed a learning theoretic frame-work to analyze the performance of contrastive objectives. However, without strong assumptions on intra-class concentration they note that contrastive objectives are fundamentally limited in the representations they are able to learn. RELIC explicitly enforces intra-class concentration via the invariance regularizer, ensuring that it generalizes under weaker assumptions. Unlike (Saunshi et al., 2019) who do not discuss augmentations, we incorporate augmentations into our theoretical explanation of contrastive methods.

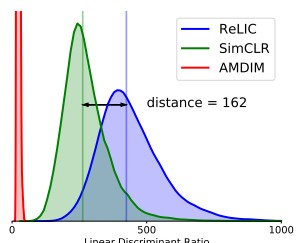

Figure 2: Distribution of the linear discriminant ratio ($F_{\mathrm{LDA}}$, see text) of $f$ for RELIC, SimCLR and AMDIM ($y$-axis clipped to aid visualization).

The reasons for the improvement in performance from AMDIM through to SimCLR and BYOL are not easily explained by either the MI maximization or the learning theoretic viewpoint. Further, it is not clear why relatively minor architectural differences between the methods result in significant differences in performance nor is it obvious how current state-of-the-art can be improved. In contrast to prior art, the performance of RELIC is explained by connections to causal theory. As such it gives a clear path for improving results by devising problem appropriate refinements, interventions and invariance penalties. Furthermore, the use of invariance penalties in RELIC as dictated by causal theory yields significantly more robust representations that generalize better than those learned with SimCLR or BYOL.

**Causality and invariance.** Recently, the notion of invariant prediction has emerged as an important operational concept in causal inference (Peters et al., 2016). This idea has been used to learn classifiers which are robust against domain shifts (Gong et al., 2016). Notably, (Heinze-Deml & Meinshausen, 2017) propose to use group structure to delineate between different environments where the aim is to minimize the classification loss while also ensuring that the conditional variance of the prediction function within each group remains small. Unlike (Heinze-Deml & Meinshausen, 2017) who use supervised data and rely on having a grouping in the training data, our approach does not rely on ground-truth targets and can flexibly create groupings of the training data if none are present. Further, we enforce invariant prediction within the group by constraining the distance between distributions resulting from contrasting data across groups.

The relationship between causal inference and semi-supervised learning has been explored in (Schölkopf et al., 2012). In particular, in order for *unlabeled* data to be helpful for learning, the relationship between the predictors and targets must be anti-causal. Our setting is somewhat different as we assume a latent decomposition of the observed variables and learn representations based on proxy targets which are refinements of the true targets.

Our invariance penalty is similar in practise to *consistency* regularizers which have recently gained popularity in semi-supervised learning (Sajjadi et al., 2016; Xie et al., 2019; Sohn et al., 2020). These typically supplement the supervised training signal with a term which enforces similarity between the function's output on two views of data. Although recently proposed consistency methods are heuristic, they have deeper roots in the more rigorous co-training approach to semi-supervised learning from multiple views (Blum & Mitchell, 1998; Kakade & Foster, 2007; Sridharan & Kakade, 2008; McWilliams & Montana, 2012; McWilliams et al., 2013). The main conceptual difference

---

[4]The set of augmentations includes Gaussian blurring, various colour distortions, flips and random cropping.

between our approach and consistency-based methods is that we do not have access to a supervised training signal and that the RELIC penalty is motivated from the underlying causal graph.

## 5 EXPERIMENTS

We first visualize the influence of the explicit invariance constraint in RELIC on the linear separability of the learned representations. We then evaluate RELIC on a number of prediction and reinforcement learning tasks for usefulness and robustness. For the prediction tasks, we test RELIC after pretraining the representation in a self-supervised way on the training set of the ImageNet ILSVRC-2012 dataset (Russakovsky et al., 2015). We evaluate RELIC in the linear evaluation setup on ImageNet and test its robustness and out-of-distribution generalization on datasets related to ImageNet. Unlike much prior work in contrastive learning which focuses specifically on computer vision tasks, we test RELIC also in the context of learning representations for reinforcement learning. Specifically, we test RELIC on the suite of Atari games (Bellemare et al., 2013) which consists of 57 diverse games of varying difficulty.

**Linear evaluation.** In order to understand how representations learned by RELIC differ from other methods, we compare it against those learned by AMDIM and SimCLR in terms of Fischer's *linear discriminant ratio* (Friedman et al., 2009): $F_{\text{LDA}} = \|\mu_k - \mu_{k'}\|^2 / \sum_{i,j \in C_k} \|f(x_i) - f(x_j)\|^2$ where $\mu_k = \frac{1}{|C_k|} \sum_{i \in C_k} f(x_i)$ is the mean of the representations of class $k$ and $C_k$ is the index set of that class. A larger $F_{\text{LDA}}$ implies that classes are more easily separated with a linear classifier. This can be achieved by either increasing distances between classes (numerator) or shrinking within-class variance (denominator).

Figure 2 shows the distribution of $F_{\text{LDA}}$ for RELIC, SimCLR and AMDIM after training as measured on the (downsampled) ImageNet validation set. The distance between medians of RELIC and SimCLR is 162. AMDIM is tightly concentrated close to 20. The invariance penalty ensures that—even though labels are *a-priori* unknown—for RELIC within-class variability of $f$ is concentrated leading to better linear separability between classes in the downstream task of interest. This is reflected in the rightward shift of the distribution of $F_{\text{LDA}}$ in Figure 2 for RELIC compared with SimCLR and AMDIM which do not impose such a constraint.

Next we evaluate RELIC's representation by training a linear classifier of top of the fixed encoder following the procedure in (Kolesnikov et al., 2019; Chen et al., 2020a) and Appendix E.4. In Table 1, we report top-1 and top-5 accuracy on the ImageNet test set. Methods denoted with * use SimCLR augmentations (Chen et al., 2020a), while methods denoted † use custom, stronger augmentations. Comparing methods which use SimCLR augmentations, RELIC outperforms competing approaches on both ResNet-50 and ResNet-50 with target network. For completeness, we report results for SwAV (Caron et al., 2020) and InfoMin (Tian et al., 2020), but note that these methods use stronger augmentations which alone have been shown to boost performance by over 5%. A fair comparison between different objectives can only be achieved under the same architecture and the same set of augmentations.

Table 1: Accuracy (in %) under linear evaluation on ImageNet for different self-supervised representation learning methods. Methods with * use SimCLR augmentations. Methods with † use custom, stronger augmentations.

| Method | | Top-1 | Top-5 |
|---|---|---|---|
| *ResNet-50 architecture* | | | |
| PIRL | | 63.6 | - |
| CPC v2 | | 63.8 | 85.3 |
| CMC | | 66.2 | 87.0 |
| SimCLR (Chen et al., 2020a) | * | 69.3 | 89.0 |
| SwAV (Caron et al., 2020) | * | 70.1 | - |
| RELIC (ours) | * | 70.3 | 89.5 |
| InfoMin Aug. (Tian et al., 2020) | † | 73.0 | 91.1 |
| SwAV (Caron et al., 2020) | † | 75.3 | - |
| *ResNet-50 with target network* | | | |
| MoCo v2 (Chen et al., 2020b) | | 71.1 | - |
| BYOL (Grill et al., 2020) | * | 74.3 | 91.6 |
| RELIC (ours) | * | 74.8 | 92.2 |

**Robustness and generalization.** We evaluate robustness and out-of-distribution generalization of RELIC's representation on datasets Imagenet-C (Hendrycks & Dietterich, 2019) and ImageNet-R (Hendrycks et al., 2020), respectively. To evaluate RELIC's representation, we train a linear classifier on top of the frozen representation following the procedure described in (Chen et al., 2020a) and appendix E.5.2. For Imagenet-C we report the mean Corruption Error (mCE) and Corruption Errors for Noise corruptions in Table 3. RELIC has significantly lower mCE than both the supervised ResNet-50 baseline and the unsupervised methods SimCLR and BYOL. Also, it has the lowest Corruption Error on 14 out of 15 corruptions when compared to SimCLR and BYOL. Thus, we see that RELIC learns the most robust representation. RELIC also outperforms SimCLR and BYOL on ImageNet-R showing its superior out-of-distribution generalization ability; see Table 2. For further details and results please consult E.5.

Table 2: Top-1 error rates for different self-supervised representation learning methods on ImageNet-R. All models are trained only on clean ImageNet images and $\text{RELIC}_T$ refers to RELIC using a ResNet-50 with target network as in BYOL (Grill et al., 2020).

| Method | Supervised | SimCLR | RELIC | BYOL | $\text{RELIC}_T$ |
|---|---|---|---|---|---|
| Top-1 Error (%) | 63.9 | 81.7 | 77.4 | 77.0 | 76.2 |

Table 3: Mean Corruption Error (mCE), mean relative Corruption Error (mrCE) and Corruption Errors for the "Noise" class of corruptions (Gaussian, Shot, Impulse) on ImageNet-C. The mCE value is the average across 75 different corruptions. Methods are trained only on clean ImageNet images.

| Method | mCE | mrCE | Gaussian | Shot | Impulse |
|---|---|---|---|---|---|
| Supervised | 76.7 | 105.0 | 80.0 | 82.0 | 83.0 |
| *ResNet-50 architecture:* | | | | | |
| SimCLR | 87.5 | 111.9 | 79.4 | 81.9 | 89.6 |
| ReLIC | 76.4 | **87.7** | 67.8 | 70.7 | 77.0 |
| *ResNet-50 with target network:* | | | | | |
| BYOL | 72.3 | 90.0 | 65.9 | 68.4 | 73.7 |
| ReLIC | **70.8** | 88.4 | **63.6** | **65.7** | **69.2** |

**Reinforcement Learning.** Much prior work in contrastive learning has focused specifically on computer vision tasks. In order to compare these approaches in a different domain, we investigate representation learning in the context of reinforcement learning. We compare RELIC as an auxiliary loss against other state of the art self-supervised losses on an agent trained on 57 Atari games. Using human normalized scores as a metric, we use the original architecture and hyperparameters of the R2D2 agent (Kapturowski et al., 2019) and supplement it with a second encoder trained with a given representation learning loss. When auxiliary losses are present, the Q-Network takes the output of the second encoder as an input. The Q-Network and the encoder are trained with separate optimizers. For the augmentation baseline, the Q-Network takes two identical encoders trained end-to-end. Table 4 shows a comparison between RELIC, SimCLR, BYOL, CURL (Srinivas et al., 2020), and feeding augmented observations directly to the agent (Kostrikov et al., 2020). We find that RELIC has a significant advantage over competing self-supervised methods, performing best in 25 out of 57 games. The next best performing method, CURL performs best in 11 games. Note that none of these methods outperform R2D2 (Kapturowski et al., 2019) which achieves superhuman performance in 52 games. Although previously published work shows the auxiliary tasks help on Atari, this is in the very low-data regime. Here we show that ReLIC is able to close the gap in the more common Atari set up. We hypothesize that the performance penalty resulting from adding auxiliary self-supervised losses stems from the naive combining of the outputs of the two encoders; we will explore alternative options for combining encoder outputs in future. Full details are presented in Section E.6.

## 6 CONCLUSION

In this work we have analyzed self-supervised learning using a causal framework. Using a causal graph, we have formalized the problem of self-supervised representation learning and derived

Table 4: Human Normalized Scores of Auxiliary Methods over 57 Atari Games.

| Atari Performance | RELIC | SimCLR | CURL | BYOL | Augmentation |
|---|---|---|---|---|---|
| Capped mean | **91.46** | 88.76 | 90.72 | 89.43 | 80.60 |
| Number of superhuman games | **51** | 49 | 49 | 49 | 34 |
| Mean | **3003.73** | 2086.16 | 2413.12 | 1769.43 | 503.15 |
| Median | **832.50** | 592.83 | 819.56 | 483.39 | 132.17 |
| 40% Percentile | 356.27 | 266.07 | **409.46** | 224.80 | 94.35 |
| 30% Percentile | **202.49** | 174.19 | 190.96 | 150.21 | 80.04 |
| 20% Percentile | **133.93** | 120.84 | 126.10 | 118.36 | 57.95 |
| 10% Percentile | **83.79** | 37.19 | 59.09 | 44.14 | 32.74 |
| 5% Percentile | **20.87** | 12.74 | 20.56 | 7.75 | 2.85 |

properties of the optimal representation. We have shown that representations need to be invariant predictors of proxy targets under interventions on features that are only correlated, but not causally related to the downstream tasks. We have leveraged data augmentations to simulate these interventions and have proposed to explicitly enforce this invariance constraint. Based on this, we have proposed a new self-supervised objective, Representation Learning via Invariant Causal Mechanisms (RELIC), that enforces invariant prediction of proxy targets across augmentations using an invariance regularizer. Further, we have generalized contrastive methods using the concept of refinements and have shown that learning a representation on refinements using the principle of invariant prediction is a sufficient condition for these representations to generalize to downstream tasks. With this, we have provided an alternative explanation to mutual information for the success of contrastive methods. Empirically we have compared RELIC against recent self-supervised methods on a variety of prediction and reinforcement learning tasks. Specifically, RELIC significantly outperforms competing methods in terms of robustness and out-of-distribution generalization of the representations it learns on ImageNet. RELIC also significantly outperforms related self-supervised methods on the Atari suite achieving superhuman performance on 51 out of 57 games. We aim to investigate the construction of more coarse-grained refinements and the empirical evaluation of different kinds of refinements in future work.

**Acknowledgements.** We thank David Balduzzi, Melanie Rey, Christina Heinze-Deml, Ilja Kuzborskij, Ali Eslami, Jeffrey de Fauw and Josip Djolonga for invaluable discussions.

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
