# OpenReview forum: "Representation Learning via Invariant Causal Mechanisms"
_ICLR.cc/2021/Conference — ICLR 2021 Poster_

### Official Review · AnonReviewer4 · 2020-10-27
**Interpretation is nice, but seems to oversimplify the problem.**

**Rating:** 6
**Confidence:** 4

**Review:**


## General Summary:

The authors propose a causal interpretation of the self-supervised representation learning problem. They introduce a method (ReLIC) that employs invariance constraints on the proxy objectives to enforce better generalization.

The invariance is enforced through an additional constraint across interventions on the style generating factor, more concretely, these interventions manifest themselves in form of simple data augmentation strategies (rotation, translation, scaling, etc).

The authors interpret contrastive learning through the concept of refinements.

To the extent of my knowledge, all relevant related work has been mentioned (Invariant Causal Prediction, Invariant Risk Minimization). It would be nice if the authors would make the separation in the main text more clear, the main difference being the regularization and the way they choose the interventions.

## Overall Recommendation

I find the method proposed in the paper novel and well explained and put in the causal framework. Overall, I think it's an accept but to be a stronger one the authors should address the points in the comments.

## Writeup

The paper is clearly written, with a few improvements that can be made (a bit of notation and some typos, unfinished sentences).


## Pros
The causal framework is well-motivated, i like the separation in content and style factors. The interpretation of self-supervised learning as invariant prediction with refinements is valid. The results seem significant, since ReLIC outperforms other approaches  in an RL (Atari) and classification setting (ImageNet).

I like the motivating sentence in the paper that real-world meta-data is abundant and can be used to construct refinements more efficiently, which speaks for the relevance of the contribution.

To the extent of my knowledge the contribution is novel, I am not aware of any other work that connected contrastive learning with the causal framework.

## Cons

The problem of choosing the intervention on style factors is a bit understated, the interventions mentioned need not be content-preserving.

I find the evaluation on the Atari benchmark a bit misguided, although the method shows superior performance in some environments, RL is exactly the setting where it is difficult to make good interventions for better generalization, i.e. it is difficult to determine in a data-driven way what separates the content from the style.

## Comments

In Section 2 would be useful to formally define $Y_t$ as a set of labels. I am not sure about the multi-environment setup, since the distribution p(X) changes also in switching environments (domain shift), which would require different formalism.

p.4 par.1 where you separate style from content. Shouldn't style be also what causes the fine-grained instance separation. For example, if I have the color of eyes of dogs. In the instance classification task, this would be part of content vs. in the cats and dogs task, this would be part of style. Which would imply that the causal graph in fig. 1  should also look differently, since color of eyes doesn't cause a dog to be a dog.

p.4 ReLIC objective. I realize the shorthand notation, but the outer expectation should be over $x ~ p(X)$?

p.4 par. 3 it feels a bit off to me to name content C a representation, whereby it's a latent causal factor that we do not observe. When talking abut representations, we mostly mean f(X)? But sure, if we would use C, we would get an invariant predictor by definition of the causal graph.

p.4. last paragraph - fine-grained problems that->than $Y_t$

p.4. last paragraph - at this point, it is very tricky to state what is a content-preserving data augmentation, this is very much tailored to the problem at hand. A very basic example, random cropping is not content preserving if it doesn't show the dog. This is going back to my critique of the work that the problem of choosing intervention is oversimplified. But I understand, given that we know how to do interventions on the style variable, we will be able to extract an invariant predictor.

p.6 par. 2  Unfinished sentence  Unlike...

p. 7. I think that the experiment on ImageNet shows exactly the point that it is necessary to find good interventions, since the methods employing strong augmentation outperform ReLIC. It would be interesting to see what is the performance of ReLIC with the same set of strong augmentations.

p. 7 table 2, why aren't the strong augmentation baselines mentioned in this table? I am not familiar with the type of augmentations that were done in those baselines.

p. 8 reinforcement learning evaluation. Again, I think this is a bit misguided. It is clear that in the general sense we need invariant representations for reinforcement learning,  but choosing the right interventions is difficult.

---

> ### Author Response · Authors · 2020-11-18
> **Response**
>
> We thank the reviewer for their positive evaluation of the paper and their comments and suggestions for improvement. We will make the suggested changes in a future version. We would like to answer specific questions here.
>
> **Problem setting:** Y are general targets these could be labels (as we make explicit in section 2) or regression targets etc. We do not consider a multi-environment setup, only different possible tasks from the same underlying p(X).
>
> **Choice of augmentations.** Part of the aim of our work is to better understand the role that augmentations play in contrastive learning. For this reason we perform like-for-like comparisons using the same set of “SimCLR” augmentations commonly used in a wide range of recent works. We do not propose new augmentations, rather we frame them in the context of interventions on a latent style variable. In the case of the SimCLR augmentations, these are broadly content preserving: it is still possible to visually determine the class of the image after augmentation.
>
> **Comparison to strong augmentations.** Methods using a stronger set of augmentations only appeared on arXiv while we were preparing this submission. We have not yet had the chance to thoroughly evaluate the augmentations proposed within because this is concurrent work. While every effort has been made to compare against all recently published and even unpublished concurrent work where possible, the experiments in question are large-scale and require a lot of time to run. That said, we will endeavour to include these comparisons in a revision or the final version. It should be noted that in [1] they note that the introduction of stronger augmentations improves the performance of SimCLR by ~2% on the Imagenet benchmark. We hypothesize that ReLIC will benefit similarly.
>
> **Using style for solving the instance discrimination problem.** While the goal of the proxy task is to achieve instance discrimination, this is normally not the final task of interest. We therefore aim to solve the instance discrimination task by learning features which are robust. In fact, as shown in Appendix B there is a trade-off between creating separation between examples using instance discrimniation and ensuring within-class concentration using our invariance regularizer.
>
> In the case of instance discrimination between dogs vs classifying dogs against cats, this does not preclude the use of eye colour as a useful feature. However, learned features are rarely this concrete: we instead expect to learn combinations of abstract features. More generally, one of the strengths of our approach is that it draws an explicit connection between the proxy tasks and the downstream tasks. However, it also suggests that if the proxy and downstream tasks are very misaligned then there is no expectation that Relic (or indeed any contrastive method) would work well.
>
> **Using C or f(X) as the representation.** The causal graph represents the idealised case. f(X) is an estimator for C. Were C known exactly we would be able to use this to solve the downstream tasks directly.
>
> **RL evaluation.** One of the key use cases for unsupervised representation learning is RL, where rewards can be sparse. We are not the first to propose using contrastive representation learning for reinforcement learning. Most notably [2] (who we compare with and out perform). It is clear that the augmentations we use are useful for learning good visual representations from pixels in order to perform the relatively simple tasks required in Atari. However, we agree that for more complex environments more thought around which interventions to use is required. This is a large open research area in its own right and we defer it for future work.
>
> [1] M. Caron, I. Misra, J. Mairal, Priya Goyal, P. Bojanowski, and Armand Joulin. (2020) Unsupervised learning of visual features by contrasting cluster assignments.
>
> [2] Aravind Srinivas, Michael Laskin, and Pieter Abbeel. (2020) Curl: Contrastive unsupervised representations for reinforcement learning.

---

### Official Review · AnonReviewer3 · 2020-10-29
**New perspective on self-supervised learning**

**Rating:** 6
**Confidence:** 4

**Review:**

In this paper, the authors propose a new understanding of self-supervised learning from a causal perspective. Specifically, a causal graph with style and content is assumed for the generating process of the inputs, such as images. Another assumption is that the down-stream tasks only rely on the content variable. By making use of the independent causal mechanism, the authors propose a new invariance regularization term, which is achieves good performance on several real datasets. Also, a new understanding of contrastive learning is provided.

Strength

The understanding of self-supervised learning from a causal perspective is novel. The causal generative model assumed in this paper seems to be reasonable in many real scenarios. Especially, the image data were generated from content and style and usually the downstream tasks such as object recognition depends on the content.

The experimental results on Imagenet and Atari demonstrate the effectiveness of the method.

Weakness

The idea of independence mechanisms was originally proposed in [1] and has deep connections to the modularity of a causal system and the concept of exogeneity in economics (Pearl, 2009). In specific, given two variables C and E, we say C is exogenous if P(E|C) remains invariant to changes in the process that generates C. In [1], the independence mechanism is defined as follows:
 “We finally assume that the mechanism is “independent” of the distribution of the cause in the sense that P(E|C) contains no information about P(C) and vice versa; in particular, if P(E|C) changes at some point in time, there is no reason to believe that P(C) changes at the same time.” When P(c) and P(E|C) both change, they change independently of each other [2]. It would be better if the authors could explore the literature a little bit more and add corresponding discussions.
[1] Schölkopf, Bernhard, et al. "On causal and anticausal learning." Proceedings of the 29th International Coference on International Conference on Machine Learning. 2012.
[2] Huang, Biwei, et al. "Causal discovery from heterogeneous/nonstationary data." Journal of Machine Learning Research 21.89 (2020): 1-53.

The proposed invariance regularization is closely related to the consistency regularization [3] in semi-supervised learning. The relation and difference to consistency regularization needs to be discussed.
[3] Sajjadi, Mehdi, Mehran Javanmardi, and Tolga Tasdizen. "Regularization with stochastic transformations and perturbations for deep semi-supervised learning." Advances in neural information processing systems. 2016.

The refinement seems counterintuitive to me. The authors define a refinement of one problem as another more fine-grained problem. For a downstream task, when could it be easier to get the constructed labels for a refinement than obtain the labels of the downstream task?

The independence of content and style seems to be a strong assumption. In computer vision, one extensively studied problem is how to make use of context, e.g., background to help object recognition. For example, a monitor will have a high probability to stay on a desk. A car has a high probability to be on the road. Could the authors give some scenarios where the context can be independent of content?

---

> ### Author Response · Authors · 2020-11-18
> **Response**
>
> We thank the reviewer for their time and constructive feedback. With respect to the mentioned references, we thank the reviewer for pointing these out and we shall add some further discussion. We would like to point out however that unlike [2] we are not doing causal discovery. We appreciate that our regularizer has the flavour of consistency regularization from [3] and elsewhere with two important differences: a) we use the KL-divergence rather than L2 because of the probabilistic interpretation of our objective  and b) our objective is completely unsupervised. We will revise the manuscript with additional discussion on this.
>
> **Constructing refinements:** as we mention in Section 3, the instance discrimination task is an example of the trivial refinement when the downstream  task is classification: the "labels" of this task are the identities of the individual data points. Therefore we can always set this up as a proxy task independently of whether we know the downstream labels or not. In other problem settings it might be the case that meta-data relating to each observation is also collected. This might be useful for constructing proxy tasks but not exactly the same as the task of interest. Other refinements might be more appropriate depending on the task which might be constructed using e.g. metadata which is routinely collected in real-world contexts (e.g. EXIF, time and location data from images).
>
> **Use of context:** this very much problem specific. There is much work in vision where the aim is to learn representations which are invariant to background and surroundings (e.g. context). Whilst a car has a high probability of being in the road, the edge cases when cars appear off-road would be an important scenario to get right since it might signify an accident. The underlying assumption is that style is not always an accurate predictor and so learning representations invariant to style will ultimately lead to more robust predictions. Our results on ImageNet-C and -R highlight this.

---

### Official Review · AnonReviewer2 · 2020-11-02
**Interesting ideas,  good problem formulation and results**

**Rating:** 7
**Confidence:** 4

**Review:**

This paper proposes a framework for self-supervised representation learning using causality. The proposed model is formulated by assuming that the Data generation schema is composed of two independent mechanisms (ie., Style and Content) and only content is relevant for learning the underlying task. Thus, the Content is a good representation of the data and the goal of representation learning could be cast as a content estimation. Then, the authors use interventions on the Style (i.e., data augmentation in their formulation) to learn invariant representation under data augmentation (Style variable). To achieve this invariant prediction they propose a new constructive objective (ReLIC).

The paper is well written and easy to follow. My only concern is that the whole proposal relies on the assumption that Data generation is composed of two independent mechanisms (S and C) and the authors utilize various data augmentations as interventions on the Style variable S as they don’t have access to S. However, no details are given for the impact of the used data augmentation techniques on the learning better representations.

---

> ### Author Response · Authors · 2020-11-18
> **Response**
>
> We thank the reviewer for their positive evaluation of the paper and their comments and suggestions for improvement.
>
> The assumption of two independent variables (content and style) generating the data is common in the literature, e.g. [1, 2]. This assumption is also often implicitly made in a lot of the computer vision literature that relies on data augmentations for training. Note that data augmentations are required to achieve state-of-the-art performance for both supervised and unsupervised recognition tasks. We make use of this assumption because it has already been proposed in other settings and also provides a useful model for the success of data augmentations in a very wide range of scenarios.
>
> For data augmentations, we use the data augmentations proposed in [3] and now widely adopted across self-supervised representation learning methods, e.g. [4]. As mentioned in our paper (on page 4) choosing a set of augmentations implicitly defines which aspects of the data are considered style and content in relation to the downstream task. So with some high-level knowledge of the downstream task (e.g. is it image classification) one can create data augmentations by combining simple transformations such as cropping, horizontal flipping, color distortion and blurring (for a full description of the augmentations used please consult [3] or appendix E.1 in our paper). Only very recently (last few months) has the use of other data augmentation schemas been considered and we plan to test our method also with these data augmentations.
>
>
> [1] Heinze-Deml and Meinshausen, (2019). Conditional Variance Penalties and Domain Shift Robustness
>
> [2] Gong et al. (2016). Domain Adaptation with Conditional Transferable Components
>
> [3] Chen et al. (2020). A Simple Framework for Contrastive Learning of Visual Representations
>
> [4] Grill et al. (2020). Bootstrap your own latent: A new approach to self-supervised Learning

---

### Official Review · AnonReviewer5 · 2020-11-06
**Strong results, but problems in the formulation**

**Rating:** 5
**Confidence:** 3

**Review:**

## Summary

This paper takes a causal viewpoint on self-supervised contrastive representation learning. The data is modeled as being generated from two independent latent factors: style and content, where content captures all information necessary for downstream tasks, and style captures everything that is affected by training augmentations. The main contribution is a specific regularizer for self-supervised contrastive learning, motivated by the assumptions about the data generation. The learned representations are evaluated in terms of robustness, classification, and generalization performance on ImageNet and in terms of performance on the RL Atari benchmark. The proposed approach is shown to outperform even very recent competing approaches.

## Pros & Cons

+ The explicit expression of the assumptions behind the data generation process as a (causal) network. It explains the assumptions made for augmentations and proxy tasks to make sense; it also nicely subsumes previous formulations.
+ Visualization of the resulting feature space
+ Performance appears great, the evaluation seems sufficient enough.

- Proof of Theorem 1 has problems, fixing it, meant to change the causal graph. Although, this might be fixable (see questions below).
- There is in fact no causal language needed for this paper. There is no causal discovery or anything happening. It is really just: Here are my assumptions about the mechanisms of data generation and everything else follows just from statistical independence. For instance, the proposed invariance criterion can also be formulated as the distribution $p(Y_t \mid C)$ being invariant to distributional shifts of $p(S)$; and trying to gain robustness against covariate shifts is nothing new per se. Being explicit about the dependency between downstream tasks is important, but was not correctly stated in the paper.

## Questions and concerns
- for intervening in a causal graph on a root note, I don't need the "do" notation. There is no need to cut the graph. So the statements are all trivial from this perspective.

- The proof of theorem 1 has problems, I think. Namely, after the first equal, you write $p^{do(s_i)}(Y_t \mid Y^R)$, which should be $p^{do(s_i)}(Y_t \mid Y^R, f(X))$ unless $Y_t \perp f(X) \mid Y^R$. In this case, you can not progress further:

$$p^{do(s_i)}(Y_t \mid Y^R, f(X)) = p(Y_t \mid Y^R, f(X), S=s_i) \neq p(Y_t \mid Y^R, f(X)) \neq p(Y_t \mid Y^R, f(X), S=s_j) = p^{do(s_j)}(Y_t \mid Y^R, f(X))$$

This is because $Y_t \not\perp S \mid f(X), Y^R$, so you can not drop the conditioning on $S$.

However, if $Y_t \perp f(X) \mid Y^R$ holds, the proof would work again. This would be the case when $C \rightarrow Y^R \rightarrow Y_t$, i.e. the refinement task "causes" the downstream task. From the causal graph of Figure 1a, this is not the case. Although this would make sense intuitively, as the instance discrimination task needs more information than the downstream tasks, and you even state something in this direction in footnote 3. So I would ask the authors to clarify the intended causal connections between $C$, $Y^R$, and $Y_t$.

- Again on the proof, it looks like the assumption that $Y^R$ is a refinement for all tasks in $Y$ is not even needed?
- Can you phrase the concept of "refinements" in terms of causality? What does it mean for task $Y^R$ to be a refinement of $Y_t$?. Although you state that you use the "causal concept of refinement", a causal explanation is not given as far as I can see.

- You change the definition of the $p^{do(a) }$ to one with two interventions $p^{do(a_{ik})}$. This suddenly appears below Eq 2. and is then used in what follows. To me, this looks like one of the most important contributions. Clearly, it does not follow from the causal graph or so directly. However, it is one way of defining the regularizer that is consistent with the framework and it seems to be important. So please improve the presentation and introduce it properly.

- How would you place downstream tasks like object detection or segmentation in your framework? I would imagine that learning for instance discrimination does not keep all the information necessary to solve these kinds of tasks.

## Suggestions and Comments
- I like that you make the modeling assumptions explicit
- Get rid of most of the causality jargon, and do not try to oversell your paper into the hyped field. Your paper is not doing causal inference or anything of the kind.
- Spend much more time on the regularizer and its definition with the 4 interventions that just fall from the sky. This is actually your contribution: defining the regularizer in the way you do it. A comparison to a simple baseline with just the X-entropy between two interventions would be good (or is that one of the baseline methods?).
- your style factors are also called nuisance factors in the literature
- consider making it more prominent in the paper that you solve the instance discrimination task
- the causal graph presented (in figure 1a: the meaning of arrows is not clearly defined)
- if the arrows are causal than it seems to contradict the refinement idea
- Fig 1b: make the 4 interventions more explicitly visible. Add the Bear picture to the top and bottom row

- also, it would be beneficial to see the experiments that support and visualize theorem 1 (e.g. to show that the KL for other tasks is also decreasing when we decrease the KL for instance discrimination task).

- Typos: signal -> signals (line 2 abstract),
- Theorem 1: probl also a quantifier for $t$ is needed

- Page 5: "and so the left hand side of 4" <- this should probably be "the right hand side of 4"

Overall: the paper has great results and can be a valuable contribution, but it has problems in the formulation and is overselling on being causal.

Update: Thanks for fixing the statement of the assumptions such that Theorem 1 can hold.   Update 4 -> 5. Some of my concerns are still not addressed in the revised version.

---

> ### Author Response · Authors · 2020-11-24
> **Response**
>
> We thank the reviewer for their comments and suggestions.
>
> *Regarding Theorem 1* -- We appreciate that the reviewer has taken a detailed look at our theoretical contribution. The proof as given in Appendix D.2 on page 14 is correct. The first equality *is correct* as $Y^R$ is a refinement of $Y_{t}$ and as such $p(Y_{t}\vert Y^{R})=p(Y_{t}\vert Y^{R}, f(X))$; in fact this independence is our motivation for considering refinements in the first place. This equality is a direct consequence of the construction of refinements and can be seen from Lemma 2. To see this note that discrete random variables (e.g. $Y^{R}$ and $Y_{t}$) induce equivalence relationships on the sample space, i.e. their events partition the sample space into equivalence classes. Since $Y^{R}$ is a refinement of $Y_{t}$, we know that the equivalence relation induced by $Y^{R}$ is finer than the equivalence relation induced by $Y_{t}$. By Lemma 2, we then know that every equivalence class induced under $Y_{t}$ can be constructed from a union of equivalence classes induced under $Y^{R}$. Thus, we have that $Y_{t}$ is a function of $Y^{R}$, i.e. $Y_{t} = g(Y^{R})$ for some $g$ and thus we have that $p(Y_{t}\vert Y^{R})=p(Y_{t}\vert Y^{R}, f(X))$. We will add this clarification in the proof of Theorem 1. We understand that the arrow from $Y_{t}$ to $Y^{R}$ in the causal graph in Figure 1 is potentially confusing as the relationship between $Y_{t}$ and $Y^{R}$ is not causal in nature. We will correct this.
>
> *Regarding the causal language of the paper* -- The language of causality is a very useful tool even outside of the domains of causal discovery or causal inference as we show in this paper. First, we use the language of causality to succinctly formalize assumptions about the data generation process. Using the resulting causal graph, by the principle of independence of cause and mechanism (a causal concept) we define the notion of an invariant representation (c.f. Equation 1). This allows us to formulate the invariant prediction criterion for learning (i.e. that $p(Y_{t}\vert C)$ is invariant to distribution shifts of $p(S)$ c.f. Equation 2) which is the cornerstone of our proposed method and motivates our objective. While we could have arrived at our proposed objective through some heuristic (as the reviewer recommends), we believe that having a solid theoretical grounding for our method and objective using causality is a much better approach as it not only provides theoretical guarantees (c.f. Theorem 1), but also enhances our understanding of the problem and opens the door for further improvements.
>
> Further comments:
> * Thank you for pointing out typos; we will fix them.
> * The do notation is used to signify the intervention performed on the style variable and its use does not depend on whether the intervened upon node has any parents.
> * As we discuss in the paragraph on lines 139-147, for the instance discrimination task, pairs of points (x_{i}, x_{j}) are needed and we note that the augmentations that need to be applied to the data also follow the same structure, i.e. they are given as a pair of augmentations (a_{l}, a_{k}). We will improve the notation in Equation 2 to a general set of augmentations so that product sets of augmentations are covered under this notation.
> * As we comment on in the paper (c.f. Page 4), the choice of data augmentations implicitly defines which aspects of the data are designated as style and content. Note that image classification, object detection and segmentation are highly related tasks (evidenced by good transfer performance between them and the use of common architectures for all these tasks). As such the information needed for object detection and segmentation is to a large degree overlapping with the information needed for image classification. In addition to that, for these tasks, we can also add more task-specific augmentations, see for example Purushwalkam and Gupta “Demystifying Contrastive Self-Supervised Learning: Invariances, Augmentations and Dataset Biases”. We will add more discussion on this in our paper.
> * Refinements have been proposed in the area of causal feature learning and for this reason we call refinements a causal concept. Please consult Chalupka et. al. “Causal feature learning: an overview” for more details on refinements.
> * Regarding comparison to cross-entropy baseline -- Note that we compare against this baseline in our experiments as this baseline corresponds to the SimCLR method.
> * Visualization of Theorem 1 - We visualize the consequences of Theorem 1 in Figure 2.

---

### Decision · Program_Chairs · 2021-01-07
**Final Decision**

**Decision:**

Accept (Poster)

**Comment:**

The manuscript proposes a causal interpretation of the self-supervised representation learning problem. The data is modeled as being generated from two independent latent factors: style and content, where content captures all information necessary for downstream tasks, and style captures everything that is affected by data augmentations (e.g. rotation, grayscaling, translation, cropping). The main contribution is a specific regularizer for self-supervised contrastive learning, motivated by the assumptions about the data generation.

Reviewers agreed that the manuscript is oversold on the causal jargon, as was noted, the manuscript does not perform any causal inference. Nevertheless, they think that there is an interesting interpretation of self-supervised learning and the results are noteworthy.